# Nutrient stress dramatically increases malaria parasite *clag2* copy number to increase host cell permeability and enable pathogen survival

Nicole B. Potchen[1☯¤], Tatiane Macedo Silva[1☯], Inderjeet Kalia[1☯], Justin B. Lack[2], Sanjay A. Desai [1]*

1 Laboratory of Malaria and Vector Research, National Institute of Allergy and Infectious Diseases, Division of Intramural Research, National Institutes of Health, Rockville, Maryland, United States of America, 2 Integrated Data Sciences Section, National Institute of Allergy and Infectious Diseases, Division of Intramural Research, National Institutes of Health, Rockville, Maryland, United States of America

☯ These authors contributed equally to this work.
¤ Current Address: Vaccine and Infectious Disease Division, Fred Hutchinson Cancer Center, Seattle, Washington, USA
* sdesai@niaid.nih.gov

## Abstract

To grow and replicate in erythrocytes, malaria parasites must increase the host cell's permeability to a broad range of nutrients. The plasmodial surface anion channel (PSAC) mediates this increased permeability and has been linked to CLAG3, a protein encoded by a multigene family conserved in *Plasmodium spp.* Surprisingly, an CLAG3 knockout parasite produced in *P. falciparum* exhibits incomplete reductions in PSAC activity, propagates normally in standard nutrient-rich media, but is unable to expand in modified media with more physiological levels of key nutrients. To explore these unexpected findings, we used *in vitro* selections on a CLAG3-null parasite and obtained a mutant capable of expansion under nutrient-limiting conditions. This growth was associated with restored solute uptake despite absence of CLAG3 protein. The mutant parasite expressed channels with characteristics of PSAC though with altered solute selectivity and lack of protease susceptibility, suggesting a modified channel and genome-level changes in the pathogen. Whole-genome sequencing revealed a dramatically increased *clag2* copy number without other relevant changes. Quantitative PCR and DNA transfection confirmed increased production of the *clag2* gene product. These findings implicate CLAG2 in direct formation of nutrient channels, suggest a new model that accounts for variable expansion of *clag* genes in *Plasmodium spp.*, and uncover a dramatic genome plasticity available to malaria parasites.

**Data availability statement:** All relevant data are within the manuscript and its Supporting Information files. The raw sequencing data generated for this study have been deposited in the NCBI Sequence Read Archive (SRA) under BioProject accession PRJNA1327946.

**Funding:** This work was funded by the Division of Intramural Research, NIAID/NIH. N.P. was a recipient of an Intramural NIAID Research Opportunities (INRO) Training Award. The funders had no role in study design, data collection and analysis, decision to publish, or preparation of the manuscript.

**Competing interests:** The authors have declared that no competing interests exist.

## Author summary

Malaria parasites grow within circulating red blood cells and acquire nutrients from human and animal plasma via a pore they insert in the host membrane. This pore is linked to CLAG3, a protein conserved in all examined malaria parasites. Surprisingly, deletion of CLAG3 only partially reduces formation of the nutrient pores, allowing parasites to grow normally under standard culture conditions that provide high levels of nutrients. This CLAG3-null parasite could not grow in modified media with two nutrients reduced to levels resembling those in human plasma. Here, we used prolonged culture of the CLAG3-null parasite in nutrient-limited medium to produce a mutant that can grow at near-normal rates. Despite its inability to express CLAG3, this mutant increased its nutrient uptake using pores with altered properties. Molecular studies revealed DNA-level amplification of the gene encoding CLAG2, a closely related protein from another parasite chromosome. Our findings suggest that malaria parasites can change their DNA in response to sustained changes in nutrient availability as may occur with host migration, climate change, or introduction into new host species.

## Introduction

Malaria parasites evade immune attack by replicating within erythrocytes of their vertebrate hosts. *P. falciparum*, a virulent and deadly human pathogen, dramatically remodels its host erythrocyte through exported effector proteins that alter the cell's deformability, cytoadherence and permeability [1–4]. These changes compromise immune evasion because surface display of parasite antigens and altered cellular properties can elicit host immune responses [5–7]. Epigenetic switching of multigene families encoding surface-displayed antigens, such as the PfEMP1 cytoadherence receptor or the RIFIN immunity modulator [8,9], is generally thought to be the parasite's primary defense against immune recognition.

The *clag* multigene family is unique because its encoded proteins represent the only surface-displayed antigens conserved throughout the genus *Plasmodium*. In contrast to other surface exposed antigens, the CLAG proteins do not function in cytoadherence or immune evasion, but instead enable nutrient uptake via formation of the plasmodial surface anion channel (PSAC) [10–14]. While the two copies on chromosome 3 (*clag3.1* and *clag3.2*) are under epigenetic control in *P. falciparum* [15,16], switching between these paralogs does not alter channel properties but it may finetune parasite replication in human infections [17]. Remarkably, a CLAG3 knockout line exhibits only partially compromised PSAC activity and expands normally in nutrient-rich RPMI 1640-based media, but cannot be propagated in media with more physiological levels of key nutrients [18]. The paralogs encoded by three *clag* genes on other chromosomes are exported into host cytosol [19], but their roles remain unclear as they have not been linked to PSAC formation [20–23]. Although not subject to mutually exclusive expression like the two *clag3* paralogs, clonal

variation and a blasticidin S resistant PSAC mutant have established epigenetic regulation of *clag2* but not of *clag8* or *clag9* in *P. falciparum* [12,15].

We now report copy number variation (CNV) as an alternate parasite strategy for modifying host cell properties. We used *in vitro* nutrient restriction on CLAG3-null parasites to select a mutant with a nearly 20-fold amplification of the *clag2* genomic locus [18]. The resulting increases in CLAG2 expression restored host cell permeability through formation of channels with altered properties. Our studies uncover a capacity for large-scale genome amplification in *Plasmodium spp.*, demonstrate that CNV is a novel strategy for altering host cell properties, reveal evolutionary forces that drive *clag* gene expansion in *Plasmodium spp.*, and implicate a CLAG2 contribution to nutrient uptake.

## Results

### Small numbers of CLAG3 knockout parasites persist and expand in nutrient-restricted media

CLAG proteins are translated in late schizont stage parasites and assemble cotranslationally with RhopH2 and RhopH3, two unrelated and conserved parasite proteins, to form the RhopH complex; this complex is targeted to rhoptry organelles in developing merozoites and transferred to the new host cell during erythrocyte invasion [19]. It is then exported into host cytosol, becomes integral at the erythrocyte membrane, and fully induces PSAC activity by the early trophozoite stage [24].

DNA transfection to study *clag3* in *P. falciparum* is complicated by the presence of two copies, *clag3.1* and *clag3.2*, that undergo epigenetic switching and monoallelic expression in most parasite clones [15]. KC5, a laboratory clone that carries a single hybrid *clag3* gene termed *clag3h* [25], was recently used to produce a *clag3*-null parasite termed *C3h-KO* [18]. We examined sorbitol and isoleucine permeabilities in synchronous trophozoite-stage cells infected with *C3h-KO* or the KC5 parent. Both solutes are primarily acquired via PSAC in infected cells [10,26]. Molecular confirmation of these transport studies has been established by knockdown of RhopH2 and RhopH3 [24]. Isoleucine uptake is critical for parasite propagation as it is absent from human hemoglobin and cannot be acquired through hemoglobin digestion [27].

Both solutes exhibited slower osmotic lysis kinetics in *C3h-KO* (Fig 1A), indicating reduced uptake. Because osmotic lysis kinetics correlate with whole-cell currents in patch-clamp [28,29], these measurements quantitatively estimate permeability reduction (Fig 1B). Sorbitol permeability was reduced by $48\pm2\%$; isoleucine exhibited a greater reduction at $68\pm2\%$ ($P<10^{-4}$, Student's *t* test, $n=7–21$ trials for each solute and parasite). It is not clear why these permeabilities are differentially affected by the loss of CLAG3, but studies implicating distinct routes through PSAC for these solutes may provide a structural basis [30].

We next examined the effect of reduced host cell permeability on *in vitro* parasite propagation and found that *C3h-KO* cultures expanded at rates comparable to KC5 in standard RPMI 1640-based medium (red vs black circles, Fig 1C), a general-purpose formulation with supraphysiological levels of most nutrients. Cultivation in a more physiological medium designed to evaluate PSAC inhibitors under conditions that simulate human plasma (PGIM, PSAC growth inhibition medium) [16], however, revealed a dramatic difference as KC5 cultures continued to expand while the *C3h-KO* parasitemia failed to increase over 8 days (Fig 1C). KC5 expansion was slower in PGIM than in the standard RPMI 1640-based medium (black triangles vs. black circles, Fig 1C), as has been reported previously for other wild-type parasite lines [16]. This reduced propagation in PGIM presumably reflects growth limitation due to nutrient restriction. Because growth rates in PGIM are comparable to those in human serum without addition of commercial media [16], PGIM may predict *in vivo* parasite nutrient access and replication more faithfully than RPMI 1640. Thus, stalled growth of the *C3h-KO* line in PGIM suggests that CLAG3 is essential for parasite expansion in human infections.

Interestingly, microscopic examination revealed occasional apparently healthy *C3h-KO* infected cells despite PGIM cultivation for more than 10 days (Fig 1D). To examine viability, we cultured *C3h-KO* in PGIM for various durations and quantified recovery from nutrient restriction upon return to the standard RPMI-based medium (S1A Fig). Each culture quickly resumed normal expansion; increasing the duration of PGIM exposure to 8 days did not increase the lag time (S1B Fig),

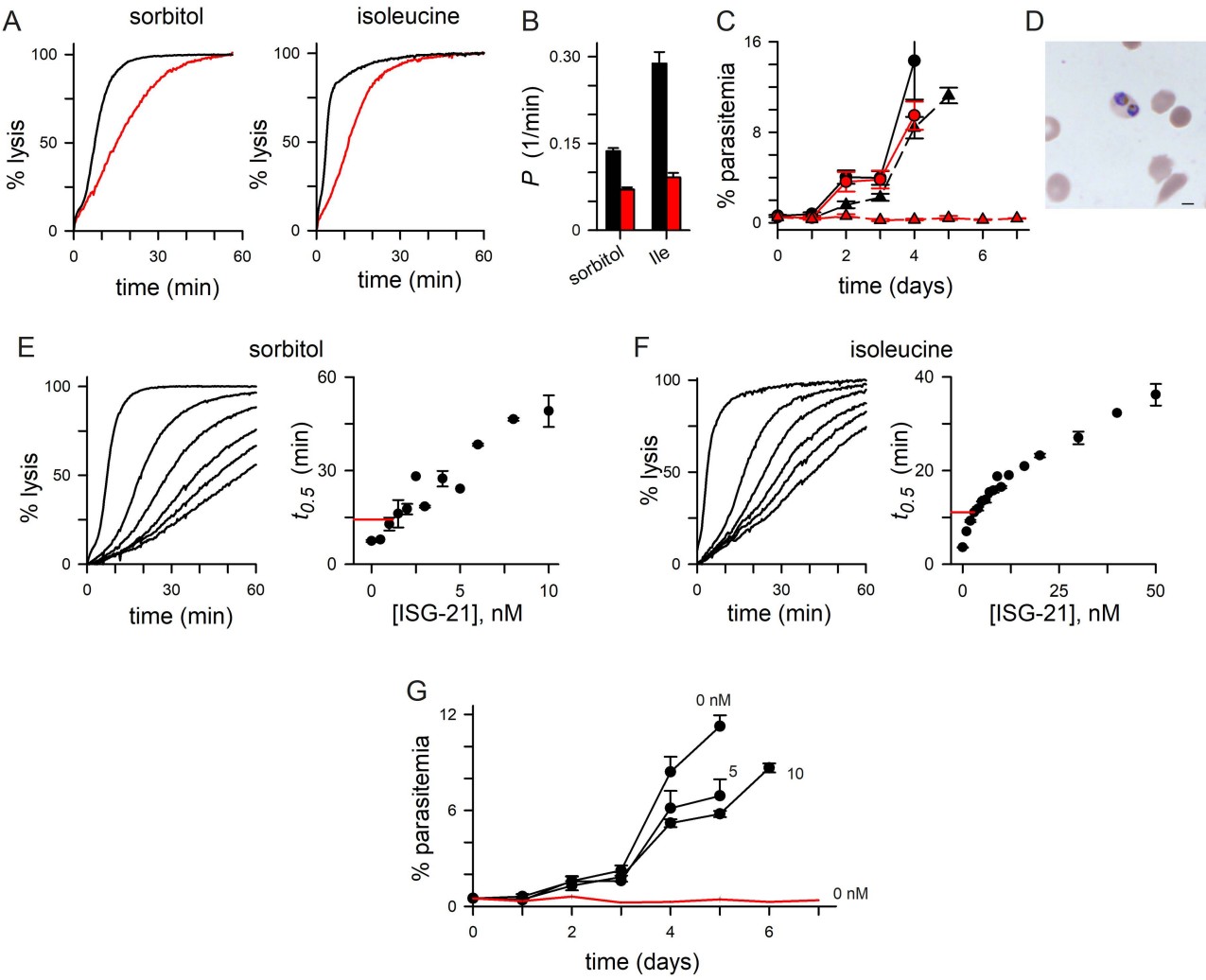

**Fig 1. Compromised growth of a CLAG3 knockout in nutrient-limited medium with rare persister parasites. A**) Osmotic lysis kinetics in indicated solutes for *C3h-KO* and the KC5 wild-type parent (red and black traces, respectively). **B**) Mean±S.E.M. permeabilities, calculated as 1/lysis halftime from osmotic lysis kinetic experiments. Black and red bars represent KC5 and *C3h-KO*. **C**) Parasitemia (% infected cells), estimated from daily microscopic examination of Giemsa-stained thin smears, for KC5 and *C3h-KO* (black and red symbols). Propagation in standard RPMI-based medium and in PGIM are shown as circles and triangles, respectively. While the knockout expands normally in RPMI-based medium, its cultures fail to expand in PGIM. Symbols, mean±S.E.M. parasitemias from 3-4 independent trials each. **D**) Microscopic image of Giemsa-stained infected cell with two trophozoite-stage *C3h-KO* parasites observed during cultivation in PGIM. Occasional ring- and schizont-stage parasites were also observed. **E**) Osmotic lysis kinetics in sorbitol with 0, 2, 4, 6, 8, 10 nM ISG-21 (left to right traces, respectively). Note slowed kinetics due to reduced sorbitol uptake by ISG-21 block. Right panel, mean±S.E.M. lysis halftimes ($t_{0.5}$) vs. ISG-21 concentration from sorbitol uptake experiments as in left panel, calculated from up to 6 independent trials at each concentration. Horizontal red line represents the mean inhibitor-free sorbitol $t_{0.5}$ for *C3h-KO* from Fig 1B. **F**) Lysis kinetics in isoleucine with 0, 10, 20, 30, 40, 50 nM ISG-21 (left to right traces). Higher concentrations of this and other inhibitors are required to block isoleucine transport through PSAC [30]. Right panel, mean±S.E.M. lysis halftimes ($t_{0.5}$) vs. ISG-21 concentration from isoleucine uptake experiments (*n* up to 3 each). Red line, mean inhibitor-free isoleucine $t_{0.5}$ for *C3h-KO* from Fig 1B. **G**) Propagation of parasites in PGIM with indicated nanomolar concentrations of ISG-21. Black symbols represent mean±S.E.M. parasitemias for wild-type; red line reflects *C3h-KO* growth, reproduced from panel C for comparison (*n*=3-4 trials each).

indicating that a stable fraction of *C3h-KO* parasites remains viable in PGIM over this duration. One possibility is that compromised nutrient acquisition, imposed here through the combination of reduced host cell permeability and limited nutrient levels in the external medium, triggers a dormant parasite state [31,32].

## Growth compromised to a greater extent than expected from permeability reduction

The marked reduction in growth using PGIM is surprising given the incomplete reduction in solute permeabilities in the *C3h-KO* line. To explore this finding quantitatively, we measured expansion of wild-type parasite cultures in the presence of ISG-21, a potent and specific PSAC inhibitor also known as MBX-2366 [33]. We quantified dose-dependent ISG-21 inhibition for both sorbitol and isoleucine uptake (Fig 1E and 1F). We then estimated the ISG-21 concentration required to reduce each solute's permeability in wild-type KC5 parasites to levels matching those in *C3h-KO* without inhibitor addition (right panels, Fig 1E and 1F). The dose-response profiles for sorbitol and isoleucine as well as the ISG-21 concentrations required to yield osmotic lysis halftimes matching values for *C3h-KO* without inhibitor addition, 1.2 and 3 nM respectively, differed modestly, possibly reflecting these solutes' distinct transport mechanisms through PSAC [30,34].

We then cultured the KC5 parental line with and without ISG-21 and compared expansion to that of *C3h-KO* cultivated without inhibitor. Remarkably, although 10 nM ISG-21 reduces permeability to a much greater extent than *clag3* knockout, growth of KC5 in PGIM despite addition of this inhibitor is comparatively well-preserved (Fig 1G), indicating that reduced permeability of these solutes does not adequately account for *C3h-KO*'s failure to expand in PGIM. These findings are consistent with parasite growth as a complex process dependent on PSAC for uptake of many nutrients from an incompletely defined medium and intracellular utilization; there may also be contributions from channel-mediated efflux of soluble metabolic waste products, some of which remain unknown.

## *In vitro* selection with PGIM restores host cell permeability via altered channels

As rare parasites were observed in nutrient-restricted cultures (Fig 1D), we used *in vitro* selection of *C3h-KO* to identify mutants capable of expansion in PGIM. A gradually increasing parasitemia suggested selection of replicating parasites (Fig 2A). After ~ 2 months, expansion yielded the *KO-PGIM* parasite pool suitable for molecular and biochemical studies. Cross-contamination with wild-type lines was excluded by PCR, which revealed an unchanged deletion locus in *KO-PGIM* and no detectable wild-type band (Fig 2B, p1-p2 amplicons); PCR from the unmodified downstream sequence also did not reveal changes that might have resulted from indels or other recombination events (*p3-p4* and *p3-p5*). DNA sequencing of the *clag3* promoter and deletion locus excluded mutations that could confer growth advantage to *KO-PGIM* and further excluded cross-contamination with wild-type parasites.

Remarkably, sorbitol permeability was significantly increased in the selected parasite when compared to *C3h-KO* (Fig 2C, *P*<0.0001, *n*=4 trials), reaching levels modestly greater than those of the KC5 wild-type parent (*P*=0.07 for comparison to KC5). To explore ongoing changes that may confer additional survival advantages, we continued *KO-PGIM* propagation in PGIM for over 1 year; transport measurements revealed modest additional increases in sorbitol permeability (Fig 2C, bar graph; 8 trials after 200 days of cultivation in PGIM), suggesting a plateau in uptake kinetics. Isoleucine permeability also increased over this period but was not fully restored to wild-type levels (Fig 2D, *P*=0.006 when compared to KC5, *n*=8–10 trials each). The correlation between restored growth in nutrient-limited PGIM and increased solute permeabilities suggests that parasite replication depends on sufficient PSAC-mediated uptake of essential solutes under physiological nutrient levels.

To explore whether the increased permeability is reversible and sustained by nutrient restriction, we next returned *KO-PGIM* to nutrient-rich RPMI 1640-based media to generate a parasite pool we termed *KO-rev*. Transport studies after continuous cultivation in nutrient-rich standard medium for up to 228 days revealed modest, statistically insignificant decreases in sorbitol and isoleucine permeabilities (Fig 2E and 2F; *P*≥0.19 for comparisons to *KO-PGIM* maintained in PGIM). This finding suggests stable increases in permeability not dependent on continued nutrient restriction.

To further evaluate the selected mutant's stability, we examined parasite expansion rates and found that *KO-rev* retained its ability to expand in PGIM despite growth in nutrient-rich RPMI 1640-based media for up to 165 days (Fig 2G). Both *KO-PGIM* and *KO-rev* could be propagated in PGIM, albeit at rates below that of the wild-type KC5 parent.

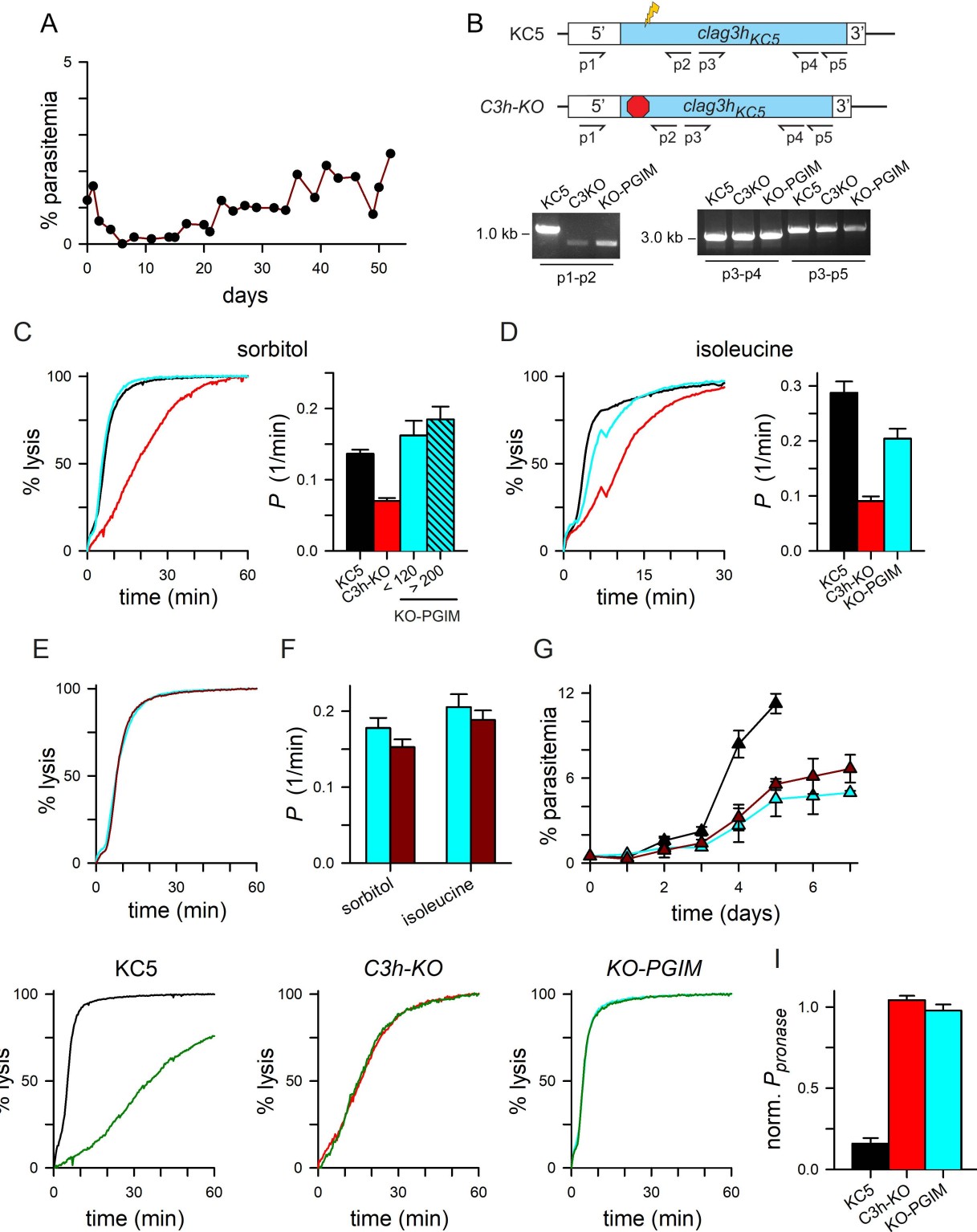

**Fig 2. Growth in PGIM after *in vitro* selection is linked to altered channel properties. A**) Timeline of parasitemia increase in a *C3h-KO* culture continuously maintained in PGIM. After 2 months, the resulting *KO-PGIM* reached levels suitable for secondary studies. **B**) Ethidium-stained gel showing PCR amplicons from indicated parasite lines and primers. Ribbon schematic above the gel shows the CRISPR/Cas9 transfection strategy used to

produce *C3h-KO*; a 359 bp smaller amplicon from primers *p1* and *p2* is predicted for *C3h-KO* and its derivatives because transfection deletes part of the *clag3* exon 1 and replaces it with three stop codons; amplicons distal to the disrupted size are unchanged. Primer sequences are provided in S1 Table. **C-D)** Osmotic lysis kinetics in sorbitol and isoleucine for the wild-type KC5 parent, the *C3h-KO* knockout, and *KO-PGIM* (black, red, and cyan traces in this and subsequent figures, respectively). Notice restored lysis kinetics for *KO-PGIM.* Bar graphs, mean ± S.E.M. sorbitol permeabilities of indicated lines. For sorbitol, *KO-PGIM* measurements are separated into trials between 68-120 days after initiation of *in vitro* PGIM selection and those after 200 days of selection; isoleucine trials for *KO-PGIM* performed between 211-330 days of selection. $n \geq 4$ trials for each condition. **E)** Lysis kinetics in sorbitol for matched cultures of *KO-PGIM* maintained in PGIM (cyan trace) or returned to nutrient-rich RPMI 1640 medium for 204 days (*KO-rev,* dark red trace). **F)** Mean ± S.E.M. permeabilities of indicated solutes for *KO-PGIM* and *KO-rev* (cyan and dark red bars); $n = 3$-12 trials each. **G)** Mean ± S.E.M. propagation of KC5, *KO-PGIM* and *KO-rev* in the nutrient-limited PGIM (black, cyan, and dark red symbols, respectively); $n = 2$-4 trials each. **H)** Osmotic lysis kinetics in sorbitol for indicated parasites with and without pronase E treatment (green trace in each panel). Note that channels on *C3h-KO* and *KO-PGIM* are pronase E-resistant. **I)** Mean ± S.E.M. permeabilities after pronase E treatment, normalized to 1.0 for matched untreated controls.

Interestingly, our analysis revealed changes in relative solute permeabilities for the selected *KO-PGIM* mutant that could not be explained by simple upregulation of transport. While cells infected with KC5 parasites have a 2-fold greater permeability to isoleucine than to sorbitol (black bars in Fig 2C and 2D; $P < 0.0001$), *KO-PGIM* parasites produce channels with negligible differences between these solutes' permeabilities (cyan bars in Fig 2F, $P = 0.21$). This altered preference reflects changes in the channel's solute selectivity and indicates that the increased permeabilities in *KO-PGIM* do not result from an increased number of unmodified channels; channels with altered properties are instead required.

We explored channel properties further using protease susceptibility studies as protease treatment of intact infected cells is known to compromise PSAC activity [10,35]. Although proteases cleave many host and parasite proteins on infected cells, genetic mapping and molecular studies have established that CLAG3 cleavage within an exposed variant loop accounts for the transport's protease susceptibility [36]. Here, we used pronase E, a mixture of nonspecific proteases active against nearly all exposed sites, and found reduced uptake by wild-type cells (Fig 2H, KC5) [36]. In contrast, channels associated with *C3h-KO* were insensitive to pronase E. Although sorbitol permeability is restored to wild-type levels, the channels on cells infected with *KO-PGIM* were also unaffected by this treatment (Fig 2H and 2I; $P < 0.0001$, one-way ANOVA with post-hoc Tukey's multiple comparisons test; $n = 3$–8 trials each parasite). Solute uptake into cells infected with *C3h-KO* and *KO-PGIM* is resistant to pronase E, further implicating channels with altered properties.

We then used whole-cell patch-clamp to examine channel-mediated Cl⁻ flux in these clonal lines [28]. While uninfected human erythrocytes have substantial exchange-restricted Cl⁻ flux through the Cl⁻/HCO$_3$ exchanger [37], they mediate negligible conductive Cl⁻ flux and have small whole-cell currents at all membrane potentials ($V_m$) [38]. Induction of PSAC activity on the infected cells permits net Cl⁻ flux and relatively large currents in whole-cell patch-clamp, which measures net ion flow only at the plasma membrane of a single infected cells (Fig 3A, schematic). These currents have been confidently linked to PSAC through electrophysiological studies of conditional knockdown of either RhopH2 or RhopH3, where net Cl⁻ transport is abolished [24].

In our studies, KC5 exhibited larger currents at negative membrane potentials ($V_m$) than at positive values (Fig 3A), consistent with PSAC's voltage-dependent gating. Remarkably, although organic solute permeabilities are reduced in *C3h-KO*, we found that Cl⁻ conductance was not significantly decreased (Fig 3A and 3B; $P = 0.99$ for post-hoc Tukey's multiple comparison test; $n = 3$–15 cells each). Also surprisingly, *KO-PGIM* produced even larger whole-cell currents ($P < 0.0001$). The channel's voltage dependence was not affected by either *clag3* knockout or by subsequent PGIM selection (Fig 3C); a modestly greater inward rectification for *Ch3-KO* and *KO-PGIM*, apparent in the I-$V_m$ plot as smaller normalized currents at $V_m$ of +100 mV, was not significant ($P = 0.05$, one-way ANOVA). Voltage-dependent inactivation, a property previously linked to a cleavable cytoplasmic domain on PSAC [39], was also preserved in these lines (Fig 3D). Cell-to-cell variation in inactivation kinetics and magnitude, which appear to reflect variable liberation of cytoplasmic channel components during whole-cell patch-clamp [39], prevented comparison of inactivation time courses. Thus, CLAG3 knockout and *in vitro* PGIM selections produce complex and unexpected changes in channel properties.

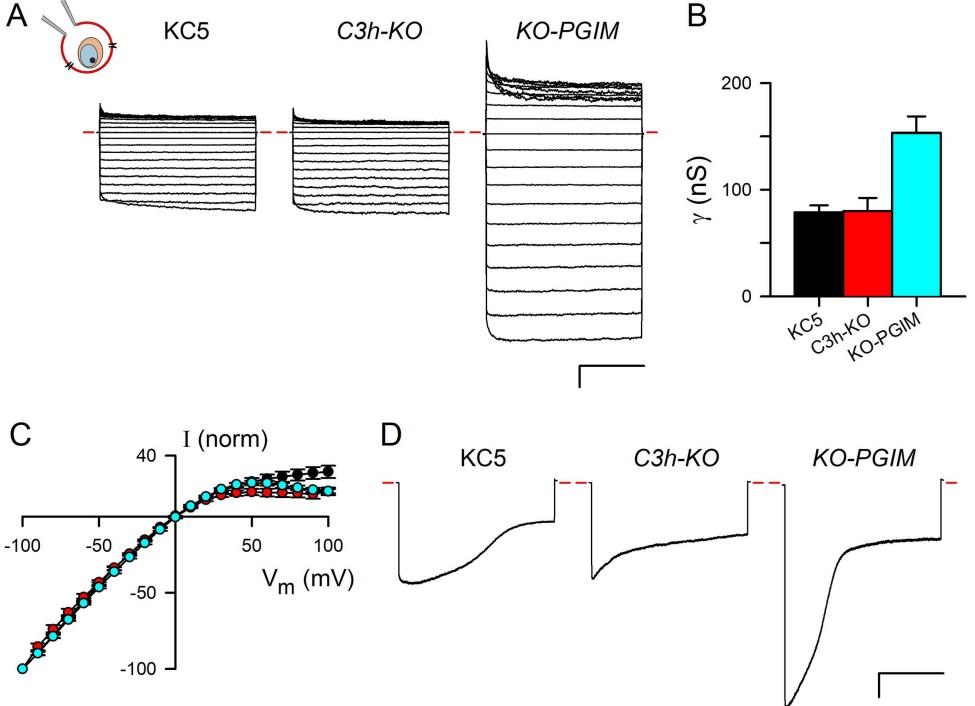

**Fig 3. Whole-cell patch-clamp reveals altered solute selectivities. A)** Whole-cell patch-clamp recordings on cells infected with indicated parasites. In each group of traces, current responses to 50 ms pulse application of membrane potentials ($V_m$) between -100 and +100 mV in 10 mV increments are superimposed; $V_m$ held at 0 mV between pulses. Red dashes, zero current level. Notice the larger negative currents at -100 mV (bottom trace in each group) than positive currents at +100 mV (top trace in each group), indicating preserved voltage-dependent gating in each clone's channels. Schematic at top left, whole-cell configuration on an infected erythrocyte. Scale bar at bottom right, 2 nA (vertical)/ 20 ms (horizontal). **B)** Mean ± S.E.M. whole-cell chord conductances (γ) determined from currents at applied $V_m$ between -100 to 0 mV ($n = 3$-15 cells for each line). **C)** Mean ± S.E.M. currents (I) at each applied $V_m$ for KC5, *C3h-KO*, and *KO-PGIM* (black, red, and cyan symbols, respectively), normalized to -100 at $V_m = $-100 mV. PSAC voltage-dependence, apparent as steeper slopes of these profiles at negative $V_m$, does not differ significantly between these clones. **D)** Voltage-dependent inactivation in response to 1 s pulses to $V_m = $-100 mV for indicated lines. Scale bar, 2 nA/400 ms.

## A markedly increased copy number for the *clag2* paralog

To examine possible molecular mechanisms for adaptation to nutrient restriction and the observed transport changes, we performed whole-genome long-read sequencing of the KC5 parent, the *C3h-KO* transfectant, and the *KO-PGIM* mutant. After quality filtering and alignment to the reference sequence, average read lengths ranged between 10,900 and 11,800 base pairs with median coverages of 128-fold to 160-fold. These long-read lengths revealed a marked increase in *clag2* copy number resulting from multiple duplications of a 6.8 kbp region that included the entire ORF and flanking untranslated sequences (Fig 4A and S2 Table).

An additional 910 bp deletion in PF3D7_1418600, one member of the multigene 5S rRNA family, was detected in *KO-PGIM* relative to KC5 and *C3h-KO*. We also compared genomic sequences to detect mutations relative to the reference and sought differences between the parental and derivative strains. Only two were found in *KO-PGIM* relative to KC5 and *C3h-KO* (blue rows, S3 Table). The PF3D7_1418600 deletion and these mutations—a 5'UTR indel in Pf3D7_0608700 and a nonsynonymous mutation in Pf3D7_0821400 encoding a small uncharacterized protein with predicted nuclear localization—were deemed unlikely to contribute to the selected mutant phenotype and were not pursued further.

PCR using a reverse primer at the start of the *clag2* gene and a forward primer at the gene's 3' end excluded detectable tandem duplication in KC5, but revealed a low-level amplification in the *C3h-KO* line (Fig 4B). This amplification, not

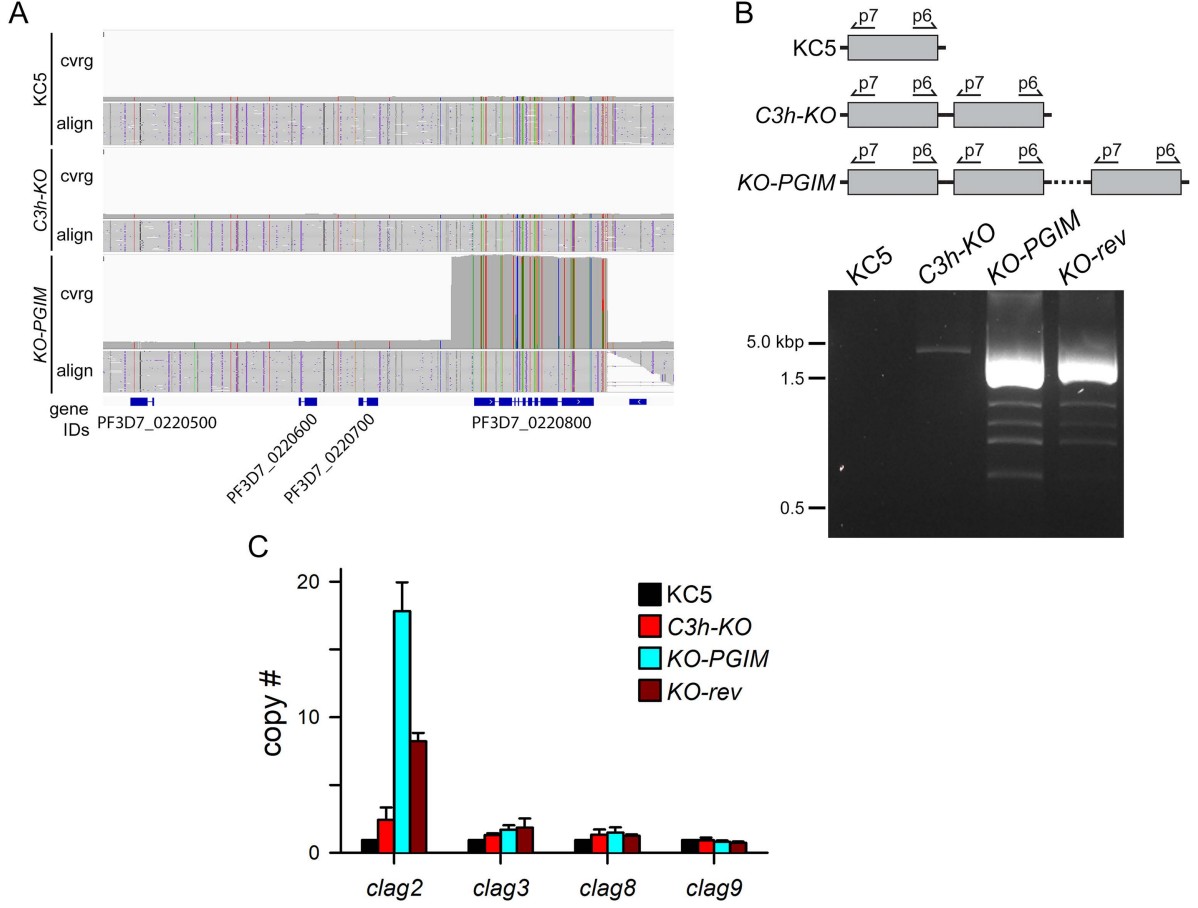

**Fig 4. Dramatic *clag2* amplification in *KO-PGIM*. A)** Integrative Genomics Viewer image of the locus containing *clag2* (PF3D7_0220800), showing sequencing coverage and alignment (cvrg and align) for KC5, *C3h-KO*, and *KO-PGIM*. Coverage is scaled to show the markedly increased coverage of a 6.8 kb region including *clag2* in *KO-PGIM*, suggesting multiple duplication events. Positions matching the reference genome sequence are gray and mismatches are colored depending upon the mutant allele. The amplified region begins 753 bp upstream of the *clag2* transcription start site and ends 15 bp before the mRNA terminus; its chromosomal positions are listed in S2 Table. **B)** Schematic shows inverted PCR strategy to identify tandem duplication events. Ethidium-stained gel showing that the wild-type KC5 does not produce an amplicon, but that *C3h-KO* produces a single band and PGIM-selected lines produce multiple bands, implicating duplication in *C3h-KO* and many copies in *KO-PGIM* and *KO-rev*. **C)** Mean ± S.E.M. DNA copy numbers of indicated genes in KC5, *C3h-KO*, *KO-PGIM* and *KO-rev* (left to right bars in each group), estimated by qPCR.

reported in previous studies of the *C3h-KO* knockout [18], may reflect a compensatory change required for growth of successfully transfected cells carrying the engineered CLAG3 knockout. PCR revealed a more dramatic *clag2* amplification in *KO-PGIM*, with several bands and more intense amplicons that parallel the multiple duplication events suggested by whole-genome sequencing. Real-time quantitative PCR (qPCR) was then used to estimate *clag2* copy numbers at 2, 18, and 8 in *C3h-KO*, *KO-PGIM*, and *KO-rev,* respectively (Fig 4C; $P < 10^{-4}$, $n = 4$ independent trials).

## CLAG2 upregulation accounts for altered channels and growth phenotype

Because *clag* genes are known to undergo complex epigenetic regulation and silencing, we next used real-time quantitative reverse transcriptase PCR (RT-qPCR) to explore whether genome-level *clag2* amplification alters transcription of *clag* paralogs, *rhoph2* and *rhoph3.* Consistent with the marked copy number increases observed in Fig 4, RT-qPCR using synchronized, trophozoite-infected parasites identified a marked upregulation of *clag2* transcription in both *KO-PGIM* and

*KO-rev* (Fig 5A; $P = 2 \times 10^{-4}$, one-way ANOVA; $n = 3$–4 trials each parasite). Significant changes in transcript abundance were not detected for other *clag* paralogs, *rhoph2*, or *rhoph3*; *clag3* transcripts remained at or below detection thresholds (Figs 5A and S2). Paralleling an apparent copy number reduction upon return to RPMI 1640-based media (Fig 4C), *clag2* expression decreased upon extended propagation in nutrient-rich media (Fig 5A, dark red bars for *KO-rev*; $P = 0.002$ in post-hoc multiple comparison test). Although evolving gene copy number, possible promoter and ORF mutations, and cell-level epigenetic marks complicate attempts to correlate copy number and transcript abundance, comparisons of our qPCR and RT-qPCR results (Figs 4C and 5A, respectively) suggest that increased *clag2* transcription results primarily from genome-level amplification without controlling effects of epigenetic marks.

To examine the effects of increased *clag2* transcription, we used CRISPR transfection of KC5, *C3h-KO* and *KO-PGIM* to introduce a C-terminal NanoLuc tag on CLAG2 (Fig 5B). Integration PCRs using the resulting limiting dilution clones—KC5$_{C2NLuc}$, *C3h-KO*$_{C2NLuc}$ and *KO-PGIM*$_{C2NLuc}$—confirmed faithful addition of the tag and replacement of all *clag2* genomic copies in each line, without residual wild-type copies of the gene (S3A Fig). Indirect immunofluorescence microscopy revealed that the modified protein was transferred to new erythrocytes after invasion and exported to the host cell cytoplasm with RhopH3 (S3B Fig), as previously reported for each CLAG paralog [19,20].

We then estimated CLAG2 abundance on nitrocellulose blots with the Nano-Glo blotting system, which can quantify NanoLuc expression [40]. ImageJ quantification with loading control normalization revealed marked increases in the tagged protein with *KO-PGIM*$_{C2NLuc}$ having greater increases than *C3h-KO*$_{C2NLuc}$ (Fig 5C right panel; $P = 0.01$, $n = 3$ independent trials each). This C-terminal tag did not compromise the increased solute uptake of the *KO-PGIM* mutant (Fig 5D). Although the transfection strategy included a *glmS* element in the 3' UTR of *clag2*, our attempts to examine conditional knockdown using this riboswitch were unsuccessful, possibly due the presence of multiple gene copies with separate promoters and glucosamine-associated toxicity on parasite cultures. Cas9-mediated cleavage during transfection, possible changes in channel-mediated nutrient uptake due to CLAG2 tagging, and variable genome remodeling in transfection subclones all may alter *clag2* copy number.

We also found that the increased CLAG2 abundance in the untransfected *KO-PGIM* could be detected in conventional immunoblots due to cross-reactivity with the anti-CLAG3 antibody (S4A Fig), consistent with significant sequence homology over the epitope used for production of this antibody (S4B Fig). We excluded low-level contamination with parasites carrying the wild-type *clag3* locus (Fig 2B); ectopic recombination to transfer the distal *clag3* sequence that encodes the antibody epitope to another *clag* paralog [10], as established for other gene families in *P. falciparum* [41], was also excluded with PCR experiments (S4C and S4D Fig).

## Discussion

Epigenetic switching of variant surface antigens is considered *P. falciparum*'s primary method of responding to changes in the host environment [42,43]. Switching and expression of single PfEMP1 paralogs, as encoded by members of the *var* gene family, allows infected cells to escape newly-mounted immune responses [44]. PfEMP1 isoforms bind to distinct endothelial receptors [45], so switching also bestows infected cell tropism for different host tissues and organs. This is particularly advantageous in pregnancy where placental sequestration leads to marked parasite expansion [46]. Epigenetic control of RIFINs, encoded by the *rif* gene family, is not under the same tight regulation as *var* gene transcription [47], but upregulation of specific *rif* genes facilitates evasion through LILRB1 or LAIR1 receptor binding to suppress host immune responses [9]. Frequent recombination between member genes for these variant antigens (50–60 *var* genes and 150–200 *rif* genes in *P. falciparum* clones) also generates distinct paralogs with new properties [48]. Despite the remarkable plasticity of these genes and their transcription profiles, studies have not revealed drastic changes in copy number for these and other variant surface antigens in *Plasmodium spp.*

We now report copy number variation (CNV) as a distinct parasite mechanism for responding to changes in host plasma. Although CNV generally cannot match the speed and reversibility characteristic of epigenetic regulation, it can

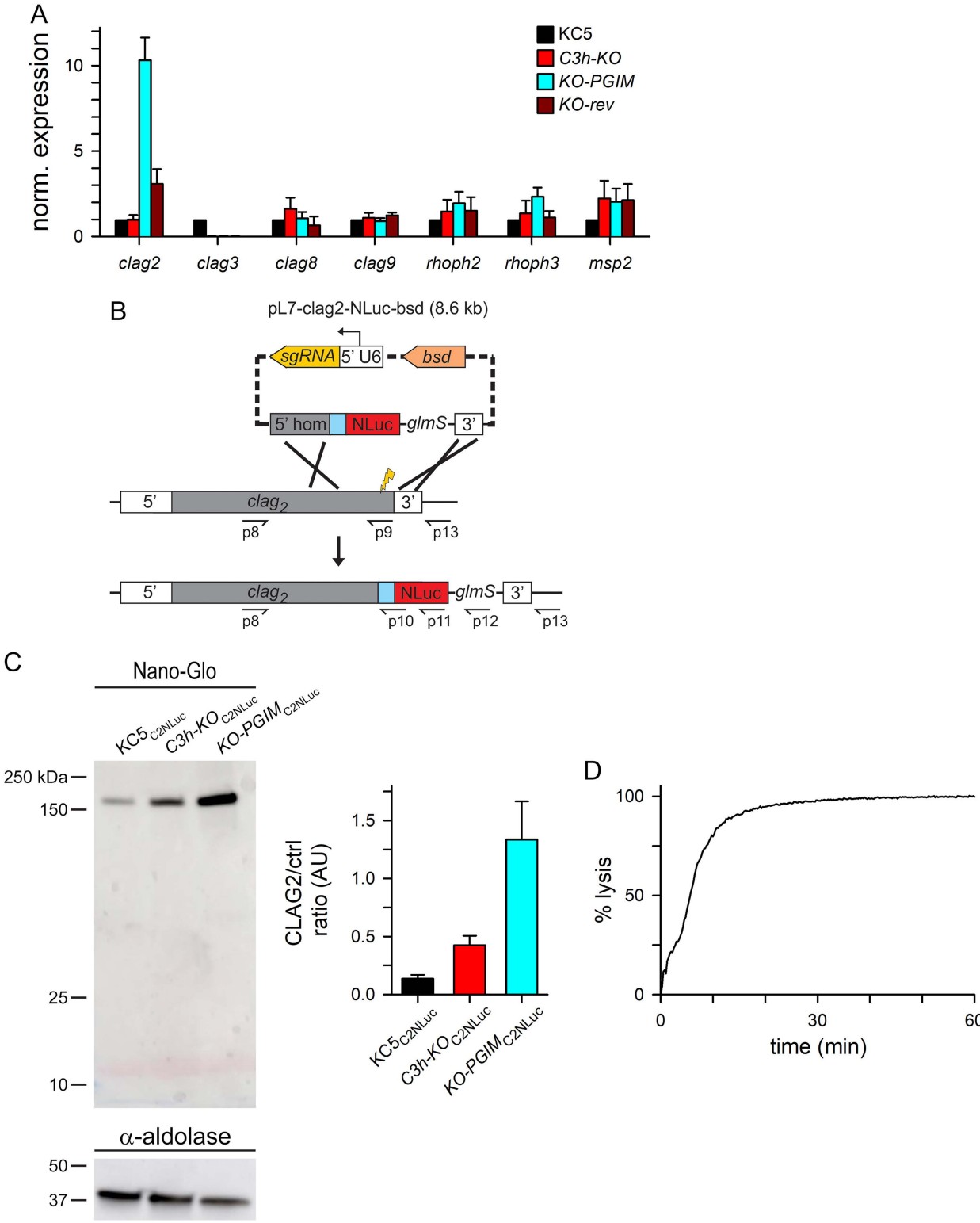

**Fig 5. Increased *clag2* transcription and translation. A)** Mean ± S.E.M. expression of indicated *clag* and *rhoph* genes in KC5, *C3h-KO*, *KO-PGIM*, and *KO-rev* parasites (black, red, cyan, and dark red bars in each group, respectively; *n* = 3-4 independent RNA harvests each), normalized to 1.0 for matched measurements using the KC5 wild-type parent. The *msp2* control is included as an unrelated gene with similar stage-specific expression;

similar *msp2* transcript levels establish matched harvest of cell cycle stages. **B)** CRISPR/Cas9 transfection strategy for C-terminal addition of a NanoLuc reporter tag on CLAG2. Indicated primers were used for PCR integration checks (S3A Fig and S1 Table). **C)** Nano-Glo blot showing the tagged CLAG2 protein in indicated lines. Loading control, anti-aldolase immunoblot at bottom. Notice the increased intensities in *C3h-KO*$_{C2NLuc}$ and *KO-PGIM*$_{C2NLuc}$. Right panel, mean ± S.E.M. CLAG2 band densities, normalized to the loading control and presented in arbitrary units (AU) for each parasite. $n = 3$ independent harvests and blots. **D)** Osmotic lysis kinetics for *KO-PGIM*$_{C2NLuc}$ in sorbitol. The lysis halftime $t_{1/2}$, 5.6 min, is comparable to untransfected *KO-PGIM*, suggesting that NanoLuc tagging does not alter channel activity. Representative of 2 independent trials.

provide adaptive advantage under sustained selective pressure. In our study, nutrient restriction appears to have promoted repeated duplication events including the full-length *clag2* gene and essential flanking sequences. The gene's subtelomeric location may have facilitated amplification as such sites are hotspots for chromosome reshuffling [49,50]. Copy number changes can arise in clinical *P. falciparum* isolates [51], but have been largely limited to intracellular parasite proteins with minimally variant sequences. PfMDR1, a transporter on the parasite's digestive vacuole, exhibits CNV to acquire resistance to mefloquine, quinine and halofantrine [52,53]; resistant clinical isolates have at most 2–4 copies of the *pfmdr1* gene, but they revert to a single copy upon drug removal to reduce the associated fitness cost [54]. GTP cyclohydrolase underwent a similar duplication in some clinical isolates to confer sulfadoxine resistance [55]. Genome-level amplifications are generally considered deleterious, especially when open reading frames are involved [56]. Thus, *clag2* amplification to 18 copies in *KO-PGIM* represents an unprecedented increase. It appears to have been necessary to restore uptake of isoleucine, an essential amino acid, establishing that nutrient restriction provided the selective pressure to drive amplification. *KO-rev*, as obtained after return of the mutant to nutrient-rich medium, exhibits a lesser copy number increase, consistent with a balance between the benefit of restored nutrient uptake and the fitness cost from high-level amplification.

There are many potential explanations for why an 18-fold increase in *clag2* copy number was required for *KO-PGIM* expansion in PGIM. These include lower transcription and translation rates for *clag2* than *clag3* (S2 Fig), differing affinities of CLAG paralogs for interaction and transport in the RhopH complex, differing efficiencies of channel subunit delivery and insertion at the host membrane (a highly regulated process for other ion channels [57]), variable subunit stoichiometries in functional channels, and differences in relative permeabilities of key nutrients in the channels formed by combinations of expressed paralogs. These possibilities may be explored when the PSAC structure is solved.

As with mutations and indels, CNV appears to rise regularly in parasite populations only to disappear quickly in the absence of associated survival advantage. CLAG3 knockout to produce *C3h-KO* provided adequate survival advantage to yield a detectable subpopulation with 2 or more copies of *clag2* despite cultivation in nutrient-rich RPMI 1640-based medium (Fig 4B); greater selective pressure upon transfer to PGIM presumably enabled growth of individual parasites carrying larger scale *clag2* amplifications. Thus, our findings help establish the pathogen's remarkable capacity for CNV under appropriate and sustained selective pressure.

When compared to other multigene copies, *clag* genes exhibit an unparalleled range of species-specific family sizes in *Plasmodium spp.* with some species such as *P. berghei* having only two copies and others exhibiting marked expansion, with the gorilla parasite *P. blacklocki* having over 35 copies [58,59]. Even within a single species, *P. falciparum* isolates are also known to have *clag* copy numbers that range between 3 and 7 even without engineered *clag3* knockout [60]. The reasons for this variable expansion are unclear, but our findings suggest nutrient restriction in specific hosts may be a key driving force. Copy number increases appear to be followed by ongoing evolution of members through mutations [60]. Interestingly, despite variable expansion, each examined *Plasmodium* species has a single *clag* closely related to the *P. falciparum clag9* gene while all other members group with *P. falciparum clag3.* While a distinct role for *clag9* has been proposed and debated [21,23], the encoded CLAG9 protein shares key phenotypes with the CLAG3 group: translation in schizonts, trafficking as a ternary complex with RhopH2 and RhopH3 via merozoite rhoptries into new erythrocytes upon invasion, and similar membrane topologies at the host cell surface [59].

In combination with copy number, epigenetic regulation of *clag* member genes serves to regulate total production of CLAG protein but the reasons for this complex regulation are unknown [61]. One possibility, switching between members to allow fine-tuning of nutrient uptake in response to changing host nutritional status [17], is appealing and consistent with some of our findings, but is unfortunately not associated with differences in nutrient permeability, as might be predicted [16]. Another possibility, switching between *clag3* genes and potential up- and down-regulation of other paralogs to permit immune evasion, is also unsatisfying, given that other multigene families require a larger number of paralogs to achieve satisfactory escape from mounted immune responses [62]. In addition to changes in PSAC-associated subunits, nutrient stress may also produce adaptive changes in the intracellular pathogen including altered expression of sugar and amino acid transporters; one example is PfAAT1, an amino acid transporter and drug susceptibility modulator localized at the parasite digestive vacuole [63].

We propose that this complex multi-layered regulation may be driven by a need to achieve adequate nutrient uptake while maintaining a sufficiently low $Na^+$ permeability to avoid premature osmotic lysis of infected cells. While PSAC has an unparalleled ability to exclude this cation [64], there is a nonzero $Na^+$ leak through the channel [65]. Upregulated nutrient permeability, as required when the host may be malnourished, must be achieved without increasing $Na^+$ uptake to levels that would lead to infected cell lysis before lifecycle completion. Fine-tuning of each paralog's expression may enable balanced permeabilities for multiple solutes in response to host nutritional status [15,20]. Consistent with this model, *in vitro* selections of mutant channel phenotypes, in the present study and in production of a mutant with reduced blasticidin S uptake [11,12,66], invariably leads to reversion to wild-type phenotypes when selection is removed.

## Methods

### Parasite culture and *in vitro* selection

Asexual *P. falciparum* parasites including KC5 and its derivative *C3h-KO*, *KO-PGIM*, and *KO-rev* lines were cultivated at 5% hematocrit in $O^+$ human erythrocytes (Interstate Blood Bank, Inc.). All cultures were maintained with regular media changes under 5% $O_2$, 5% $CO_2$, 90% $N_2$ at 37 °C. Standard, nutrient-rich propagation used RPMI 1640 medium supplemented with 25 mM HEPES, 31 mM $NaHCO_3$, 0.37 mM hypoxanthine, 10 µg/mL gentamicin, and 0.5% w/v NZ Microbio BSA (MP Biomedicals).

*In vitro* selection for *C3h-KO* parasites capable of expansion in physiological, nutrient-restricted medium was performed using PGIM medium with modifications [16]. PGIM was prepared using glutamine and isoleucine deficient RPMI 1640 (US Biological) supplemented with 25 mM HEPES, 31 mM $NaHCO_3$, 3.01 µM hypoxanthine, 11.4 µM L-isoleucine, 10 µg/mL gentamicin, and 0.5% w/v NZ Microbio BSA. Erythrocytes were washed in PGIM before addition to cultures to minimize carryover of supraphysiological nutrient levels. Propagation was evaluated with microscopic examination of Giemsa-stained smears. Expansion rate analysis used sorbitol-synchronized cultures seeded at 0.5% initial ring-stage parasitemias. Some comparisons of parasite growth rates used blinded examination of smears to exclude observer bias. Two independent trials of *in vitro* PGIM selection produced similar results.

DNA sequencing and PCR were performed with selected lines to exclude contamination by other parasite strains and to evaluate possible genomic recombination. DNA was isolated from parasite cultures using Quick-DNA Miniprep kit (Zymo Research) according to the manufacturer's protocol. Primers used are listed in S1 Table.

### Solute uptake measurements

PSAC-mediated uptake of solutes was measured using transmittance-based tracking of infected cell osmotic lysis. Synchronous cultures containing trophozoite- and schizont-stage infected cells were harvested and enriched to > 95% parasitemia using percoll-sorbitol density gradient separation. Prior studies have found that trophozoite- and schizont-infected cells have maximally induced PSAC activity on their host cells. Enriched cells were washed in saline (150 mM NaCl, 20 mM HEPES, pH 7.4) before resuspension at 0.125% hematocrit in lysis solution (280 mM sorbitol or 280 mM isoleucine

with 20 mM HEPES, 0.1 mg/mL BSA, pH 7.4) at 37 °C. Solute uptake and the resulting osmotic lysis were tracked at 15–20 s intervals by recording transmittance of 700 nm light through the cell suspension (DU800 spectrophotometer with Peltier temperature control, Beckman Coulter). Where used, ISG-21, a potent PSAC inhibitor [16], was freshly prepared in lysis solution before each experiment. Solute permeabilities and inhibitor efficacies were determined using locally developed code (SigmaPlot, Systat), based on a two-compartment model of infected cell osmotic lysis [29].

## Protein blots

CLAG protein expression was evaluated with immunoblotting of cell membranes. Sorbitol-synchronized cultures were evaluated for maturity by examination of serial Giemsa-stained smears and harvested at matched levels of trophozoite- and schizont-infected cells. Infected cells were enriched by percoll-sorbitol separation and lysed in hypotonic buffer (7.5 mM $Na_2HPO_4$, 1 mM EDTA, 1 mM PMSF at pH 7.5) at 4 °C. Membrane-associated proteins were enriched by centrifugation (14,000 x $g$, 30 min at 4 °C) and solubilized in modified Laemmli buffer with a final 6% SDS concentration. Proteins were separated by electrophoresis in 4–15% Mini-PROTEAN TGX gels (Bio-RAD) and transferred to nitrocellulose membrane. This membrane was blocked for 1 h in 3% milk solution in TBST (150 mM NaCl, 20 mM TrisHCl, pH 7.4 with 0.1% Tween20) at room temperature. Primary anti-CLAG3 and secondary HRP-conjugated anti-mouse antibodies were applied in blocking buffer as described previously [24]. Bound antibodies were visualized on Hyblot X-ray film after addition of Clarity Western ECL chemiluminescent substrate (Bio-Rad).

Nano-Glo blots (HiBiT blotting system, Promega) were used to quantify CLAG2 abundance in engineered lines expressing NanoLuc-tagged protein using antibody-free blotting to measure luminescence on membranes [40]. After SDS-PAGE and transfer to nitrocellulose membranes, blots were washed in 150 mM NaCl, 20 mM Tris-HCl (pH 7.4) with 0.1% Tween 20. Furimazine was then added in blotting buffer at a 1:500 dilution before imaging and band intensity quantification using ImageJ software (NIH) and normalization with an aldolase loading control (HRP anti-Plasmodium aldolase antibody, Abcam).

## Protease susceptibility

Enriched late-stage infected cells were washed and resuspended in PBS-2 (140 mM NaCl, 2.7 mM KCl, 9.6 mM $Na_2HPO_4$, 1.5 mM $KH_2PO_4$, 0.6 mM $CaCl_2$, 1 mM $MgCl_2$, pH 7.4) with 0.5 mg/mL pronase E (Sigma Aldrich) at 2.5% hematocrit. Cells were incubated with protease for 1 h at 37 °C. The cells were then washed with ice-cold PBS-2 supplemented with 1 mM phenylmethylsulfonyl fluoride (PMSF) and 1 mM EDTA before resuspension in saline for solute uptake measurements or hypotonic buffer for harvest of membranes and immunoblotting as described above.

## Whole-genome sequencing and analysis

The parental KC5, *C3h-KO* transfectant, and the selected *KO-PGIM P. falciparum* clones were sequenced using the PacBio SMRT Sequel II Cell with standard PacBio protocol. The samples yielded 259,683–390,238 ccs reads each, with mean read lengths of 10.9-11.8 kb and mean read qualities of QV29.7 to QV30.3. CCS reads were mapped to the PF3D7 reference genome (GCA_000002765) using pbmm2 v1.16.0 (https://github.com/PacificBiosciences/pbmm2), and resulting BAM files underwent INDEL realignment using GATK v3.8-1 [67]. Variant calling was then performed using the Haplotype-Caller from GATK v4.1.9.0 followed by joint genotyping. Resulting raw variants were then filtered with SNPs and INDELs less than 2 base pairs in length filtered at QD < 2.0 and INDELs ≥ 2 base pairs filtered at QD < 5.0, as described [68]. Variants with < 0.75 frequency in each sample were removed to minimize mapping artifacts and exclude low-quality regions or variant positions with high clonal diversity. Highly clustered variants that presumably represent mismapped, problematic regions were visually confirmed as artifacts before exclusion from downstream analysis. Structural variant calling was performed using pbsv v2.9.0 (https://github.com/PacificBiosciences/pbsv); structural variants were annotated using the PF3D7 annotation (GCA_000002765) and VEP release 111 [69]. The raw sequencing data generated for this study have been deposited in the NCBI Sequence Read Archive (SRA) under BioProject accession **PRJNA1327946**.

## Quantitative PCR for gene copy number

Genomic DNA was isolated from cultured parasites and used for *clag* gene copy number determination after assessing quality and concentration by NanoDrop (Thermo Fisher Scientific). Gene-specific primers (S1 Table) were used in dye-based qPCR (Luna Universal qPCR Master Mix, New England Biolabs) with amplification tracked in a real-time thermal cycler (Biorad CFX96). Ct values from three replicates were averaged and normalized to zero for alpha tubulin 1 (PF3D7_0903700). Copy number was estimated using the $2^{-\Delta\Delta C_T}$ method from four independent DNA harvests.

## Quantitative reverse transcriptase PCR

Gene expression was measured with real-time PCR using total RNA harvested from tightly synchronized cultures. Cultures were synchronized with two sorbitol treatments separated by 6 h. Serial smears were then used to obtain matched, schizont-stage infected cells, which were then further synchronized with percoll-sorbitol enrichment before freezing in TRIzol reagent. Total RNA was isolated using Direct-zol RNA MiniPrep kit (Zymo Research) according to the manufacturer's protocol. Isolated total RNA (800 ng) was annealed with 250 ng random hexamer primer, heated to 65 °C for five minutes and slowly cooled. First strand cDNA was synthesized with SuperScript IV Reverse Transcriptase (ThermoFisher) according to the manufacture's protocol. The resulting cDNA was diluted 10-fold and used for qRT-PCR (QuantiTect SYBR Green, Qiagen), as described previously [18]. Primers were designed to yield specific ~120 bp amplicons for indicated genes and are listed in S1 Table. qRT-PCR was carried out using the iCycler IQ multicolor real-time PCR system (Bio-Rad) programmed for denaturation at 95°C for 15 min followed by 40 cycles of annealing at 52 °C and extension at 62°C for 30 s each.

Gene expression was quantified using the $2^{-\Delta\Delta C_T}$ method [70], with *a-tubulin 1* (PF3D7_0903700) as a constitutively expressed loading control; *msp2* was included as a stage-specific control as its transcription matches the preferential expression of *clag* and *rhoph* genes at the mature schizont stage. Normalized expression is also reported by comparison to matched trials using RNA from wild-type KC5. All reactions were performed in triplicate and are presented as the mean ± S.E.M. of results from 3-4 independent RNA harvests for each transfectant or selected mutant.

## DNA transfection

CRISPR/Cas9 transfection was used to produce engineered lines that express CLAG2 with a C-terminal NanoLuc tag. A previously generated *pL7* plasmid for epitope tagging CLAG2 with a high-scoring sgRNA, 5'- ACATACTAGTAATATC CAAA-3', was modified using InFusion cloning (Clontech) to introduce a full-length NanoLuc tag after an unmodified CLAG2 protein with a two-residue linker (Bgl11 restriction site). A *bsd* selection marker cassette was also cloned into the plasmid and verified by sequencing. Cas9 was expressed from a separate *pUF1-Cas9* plasmid. After transfection of KC5, *C3h-KO*, and *KO-PGIM*, cultures were selected with 2.5 µg/mL blasticidin S and 1.5 µM DSM-1 (5-methyl[1,2, 4] triazolo[1,5-a]pyrimidin-7-yl)naphthalen-2-ylamine). *KO-PGIM* was cultivated in PGIM throughout the transfection process. After parasite outgrowth and PCR confirmation of integration, limiting dilution cloning was performed for all transfectant lines without further exposure to blasticidin S. All experiments were performed with sequence-verified clones.

## Indirect Immunofluorescence microscopy

Indirect immunofluorescence assays (IFA) were performed using freshly prepared thin blood smears after fixation with 1:1 acetone: methanol at -20 °C. After air drying, slides were blocked with 3% BSA in PBS for 1 h at RT and incubated with primary antibody in blocking buffer (rabbit anti-RhopH3 at 1:1000; mouse anti-NanoLuc antibody at 1:100, Promega) for 3 h at RT under a coverslip. After two washes, secondary antibodies (Alexa Fluor 488-conjugated goat anti-rabbit at 1:500, Invitrogen; Alexa Fluor 594-conjugated goat anti-mouse at 1:500, Invitrogen) with 1 µg/mL 4',6-diamidino-2-phenylindole (DAPI; Genetex) was added in blocking buffer and incubated for 1 h at RT. After washes, slides were dried and mounted

using ProLong Diamond Antifade Mountant (Molecular Probes). Images were collected on a Leica SP8 microscope using a 63x oil immersion objective. DAPI, Alexa Fluor 488, and Alexa Fluor 594 fluorescence were captured using excitation/emission wavelengths of 377/447 nm, 499/520 nm, and 590/618 nm, respectively. Images were processed using Leica LAS X and Huygens Essential software.

## Statistical analysis

Numerical data are shown as mean $\pm$ S.E.M of independent trials, as reported. Statistical significance was calculated using unpaired Student's *t*-test or one-way ANOVA with post-hoc testing as appropriate. Significance was accepted at $P < 0.05$.

## Supporting information

**S1 Fig. Rare *C3h-KO* parasites seen during PGIM cultivation resume expansion upon transfer to nutrient-rich medium. A)** *C3h-KO* expansion after cultivation in PGIM for 0, 2, 4, 6, or 8 days prior to return to RPMI 1640-based medium (left to right curves; symbols represent mean of 3 trials). Increasing the duration of PGIM exposure does not extend the lag before resumed growth. **B)** Time to 2% parasitemia after return to RPMI 1640-based medium for *C3h-KO* cultures seeded in PGIM for indicated number of days. Symbols represent mean $\pm$ S.E.M. estimated by linear interpolation of daily parasitemia estimates from microscopic observation; $n = 3$ trials each. Recovery time does not increase with extended PGIM exposure, indicating that these parasites remain viable.
(TIF)

**S2 Fig. Transcript abundances for *clag* and *rhoph* genes.** Mean $\pm$ S.E.M. transcription of indicated *clag* and *rhoph* genes in KC5, *C3h-KO*, *KO-PGIM*, and *KO-rev* parasites (black, red, cyan, and dark red bars in each group, respectively), calculated as $2^{(-\Delta Ct)}$ using $\Delta Ct$ relative to the α-tubulin loading control. *msp2*, unrelated gene with similar stage-specific expression. Values reflect the RT-qPCR experiments shown in Fig 5A without normalization to the wild-type control.
(TIF)

**S3 Fig. Engineered parasites carrying a NanoLuc reporter at CLAG2 C-terminus. A)** Ethidium-stained gels using indicated primers and DNA from transfectant clones and the wild-type KC5 parent. Primer pair *p8-p9* is specific for the wild-type *clag2* locus while other pairs yield amplicons upon integration. Loss of the *p8-p9* amplicon in KC5$_{C2NLuc,}$ *C3h-KO*$_{C2NLuc}$, and *KO-PGIM*$_{C2NLuc}$ indicates complete replacement of all genomic *clag2* copies with the integration cassette, which adds a C-terminal NanoLuc reporter. Expected sizes (in bp): *p8-p9*, 605; *p8-p10*, 619; *p8-p11*, 1027; *p8-p12*, 1465; *p8-p13*, 1975 in transfectant lines and 1066 in wild-type KC5. Primer sequences are in S1 Table. **B)** Indirect immunofluorescence confocal microscopy images of a trophozoite-stage *KO-PGIM*$_{C2NLuc}$ parasite probed with anti-RhopH3 and anti-NanoLuc antibodies (green and red, respectively). Scale bar, 5 μm. The NanoLuc-tagged CLAG2 protein is delivered to infected cells after invasion and is exported into host cytosol [24].
(TIF)

**S4 Fig. Increased CLAG2 production in *KO-PGIM* is detected by a weakly cross-reactive anti-CLAG3 antibody. A)** Anti-CLAG3 immunoblot using total membranes from indicated parasites. Samples used percoll-sorbitol enriched infected cells and were match-loaded. A low-intensity band is detected in *KO-PGIM* lysates but absent in the unselected *C3h-KO* parent. **B)** Sequence alignment of KC5 CLAG2 sequence with the recombinant epitope used to produce anti-CLAG3 antibody. Identical residues are highlighted in red; the ruler is numbered according to CLAG2. As the band detected in *KO-PGIM* reflects cross-reaction with the CLAG3 C-terminal epitope, we did not attempt to quantify band intensities. **C)** Ethidium-stained gels showing unaltered detection of *clag* genes in *KO-PGIM* when compared to the wild-type KC5 line. PCR of *clag8* was performed in two fragments to facilitate specific detection. Ribbon schematic at top shows the

sites recognized by each primer; primer sequences are in S1 Table. **D)** Gels showing absence of ectopic recombination between *clag3* and other paralogs in either KC5 or *KO-PGIM*.
(TIF)

**S1 Table. Primers used in this study.**
(XLSX)

**S2 Table. Structural variants in *KO-PGIM*.** Structural variants from whole-genome sequencing filtered for variants that differ between *C3h-KO* and *KO-PGIM*. The two filtered variants are tabulated with the impacted genes, chromosomal location, structural length, mutant allele, genotypes, and flanking sequencing depth.
(XLSX)

**S3 Table. Mutations and indels in *C3h-KO* and *KO-PGIM*.** Variants identified from whole-genome sequencing of parental and mutant strains and mapped to the 3D7 reference strain. Following initial quality filtering outlined in the Methods, variants were filtered that did not differ between the 3 strains. The KC5, *C3h-KO*, and *KO-PGIM* genotypes are reported as 0 when identical to the reference and 1 when a mutant allele was called. The VAR column reflects the number of parasites that differed from 3D7 at the indicated position. Columns with.DP and.GT suffixes indicate read depth and the primary allele for each parasite, respectively. Blue highlighted rows indicate high-confidence mutations in *KO-PGIM* relative to *C3h-KO*; green highlights reflect those in *C3h-KO* relative to wild-type KC5.
(XLSX)

## Acknowledgments

We thank BEI Resources for DSM-1 and Anna Crater for intellectual and technical input during early stages of these studies.

## Author contributions

**Conceptualization:** Nicole B. Potchen, Sanjay A. Desai.

**Formal analysis:** Nicole B. Potchen, Inderjeet Kalia, Justin B. Lack, Sanjay A. Desai.

**Investigation:** Nicole B. Potchen, Tatiane Macedo Silva, Inderjeet Kalia.

**Methodology:** Justin B. Lack, Sanjay A. Desai.

**Supervision:** Sanjay A. Desai.

**Writing – original draft:** Nicole B. Potchen, Sanjay A. Desai.

**Writing – review & editing:** Nicole B. Potchen, Tatiane Macedo Silva, Inderjeet Kalia, Justin B. Lack, Sanjay A. Desai.

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
