## [Decision Letter · Decision Letter 0]

7 Aug 2025

PPATHOGENS-D-25-01488

Nutrient stress dramatically increases malaria parasite clag2 copy number to increase host cell permeability and enable pathogen survival

PLOS Pathogens

Dear Dr. Desai,

Thank you for submitting your manuscript to PLOS Pathogens. After careful consideration, we feel that it has merit but does not fully meet PLOS Pathogens's publication criteria as it currently stands. Therefore, we invite you to submit a revised version of the manuscript that addresses the points raised during the review process.

Please submit your revised manuscript within 30 days Oct 06 2025 11:59PM. If you will need more time than this to complete your revisions, please reply to this message or contact the journal office at plospathogens@plos.org. Please include the following items when submitting your revised manuscript:

We look forward to receiving your revised manuscript.

Kind regards,

Tania F. de Koning-Ward

Academic Editor

PLOS Pathogens

Dominique Soldati-Favre

Section Editor

PLOS Pathogens

Sumita Bhaduri-McIntosh

Editor-in-Chief

PLOS Pathogens

orcid.org/0000-0003-2946-9497

Michael Malim

Editor-in-Chief

PLOS Pathogens

orcid.org/0000-0002-7699-2064

**Additional Editor Comments:**

Your manuscript has been reviewed by three assessors and the consensus was that the work was well executed and provided compelling evidence that amplification of the clag2 gene in P. falciparum parasites lacking CLAG3 permit growth of the parasite under nutrient-limiting conditions via the restoration of solute uptake. Whilst no new experiments are requested by the reviewers, some conclusions require a more cautious interpretation. In addition, some further clarifications and discussion points are also required. These are outlined in the reviewers comments below. Finally, the sequencing data should be deposited in a public repository.

**Journal Requirements:**

At this stage, the following Authors/Authors require contributions: Sanjay A. Desai. Please ensure that the full contributions of each author are acknowledged in the "Add/Edit/Remove Authors" section of our submission form.

3) Please ensure that the funders and grant numbers match between the Financial Disclosure field and the Funding Information tab in your submission form. Note that the funders must be provided in the same order in both places as well.

**Reviewers' Comments:**

Reviewer's Responses to Questions

**Part I - Summary**

Reviewer #1: This manuscript presents compelling and well-executed work that challenges the prevailing view of PSAC composition by demonstrating that P. falciparum parasites lacking CLAG3 can restore solute uptake and growth under nutrient-limited conditions via amplification of the clag2 gene. The authors provide strong genetic and phenotypic evidence suggesting that CLAG2 can compensate for the loss of CLAG3, leading to a revised model of channel formation and nutrient acquisition in the parasite. This finding highlights the genomic plasticity of P. falciparum and extends our understanding of how the parasite adapts to environmental stress.

However, some conclusions—specifically those regarding isoleucine uptake via PSAC—require a more cautious interpretation. Isoleucine is a bulky, hydrophobic amino acid not generally considered a primary PSAC substrate. The observed reduction in isoleucine uptake in CLAG3 knockout lines may reflect indirect effects on parasite physiology or other transport systems (e.g., amino acid transporters such as PfAAT1 (PF3D7_0629500)) rather than direct passage through PSAC. Acknowledging this limitation and refining some interpretations would strengthen the manuscript. Additionally, clarification of the parasite stage at which permeability and expression assays were performed is important, particularly in the context of possible altered IDC duration under nutrient-limited conditions.

Reviewer #2: The manuscript by Potchen et al. is a continuation of previous work from the group, in which they showed that CLAG3 proteins are an essential component of the parasite-encoded PSAC channel, involved in nutrient transport, and later demonstrated that CLAG3 is not essential for parasite growth in medium with supraphysiological nutrient concentrations (i.e., RPMI) but is essential for growth in medium with limited concentration of specific nutrients (PGIM). They also previously suggested (but not conclusively demonstrated) that other CLAG paralogs are also involved in PSAC formation.

Here the authors show that adaptation of a CLAG3-defficient P. falciparum line to grow in PGIM was associated with a compensatory amplification of the clag2 locus. This implies that CLAG2 can compensate for absence of CLAG3, providing evidence for the involvement of CLAG2 in PSAC formation. The results presented also demonstrate that CLAG2 and CLAG3 form channels with different characteristics (e.g., in terms of solute specificity).

The main conclusions are convincing and well demonstrated by the data. The manuscript is technically sound and clearly written, and the figures are also clearly presented. I only have minor suggestions to improve the manuscript, which do not require new experiments.

Reviewer #3: PSAC is a parasite-induced nutrient channel in the RBC membrane important for parasite growth under physiological conditions. CLAG3 is key component of PSAC and this study shows that loss of CLAG3 expression arrests parasite growth in physiological media.

Loss of CLAG3 reduced Ile transport 3-4 fold but did not abolish it, indicating that transport is not fully dependent on CLAG3.

Long-term maintenance of clag3-deficient parasites in physiological media can select for revertant mutants that recover the ability to grow in physiological media and the transport rate of Sorbitol/Ile is restored to parental levels.

Changes in the relative transport of sorbitol vs Ile make it clear that the channel composition must be changing.

This reversion phenotype is associated with an 18-fold increase in the copy numper of clag2.

Subsequent growth of the revertants in RPMI leads to a reduction from 18 to 9 clag2 copies.

Change in copy number correlates well with expression change at mRNA level.

Tagging of the clag2 copies with NLuc showed ~3x increased protein levels in revertant relative to the C3h-KO parent but also shows ~4x increase in C3h-KO relative to KC5 parents with the intact C3h gene.

This is a well designed and executed study that sheds new light on how parasites can acquire nutrients and the relative functions of clag2 and clag3.

**Part II – Major Issues: Key Experiments Required for Acceptance**

Reviewer #1: No new experiments are strictly required for acceptance.

However, the following points should be addressed in the text to clarify interpretations and add context to the results:

1. Refine the interpretation of isoleucine as a PSAC substrate:

The manuscript currently suggests that isoleucine is acquired via PSAC (e.g., Page 4, lines 14–16). This statement should be revised or qualified, as current knowledge does not support isoleucine as a primary PSAC substrate due to its size and hydrophobicity. The authors should consider alternative explanations for the observed decrease in isoleucine uptake in CLAG3-deficient lines.

2. Clarify parasite stage and IDC timing under PGIM conditions:

According to Subudhi et al transcriptomic data (DOI: 10.1038/s41467-020-16593-y) deposited in PlasmoDB, clag3.1 and clag3.2 are expressed primarily between 34–48 hours post-invasion. Thus, since CLAG3 expression is stage-specific, it is essential to state whether parasite stages were matched across conditions. However, the experimental timepoints (in hours post-invasion) used for permeability assays and RNA/protein analyses are not clearly stated. Also, the PGIM medium could affect the parasite’s IDC, potentially prolonging the cycle, this could influence gene expression and protein levels. The authors should specify the time post-invasion at which experimental assays were performed, and whether parasite staging was carefully matched between conditions to ensure valid comparisons.

Reviewer #2: -

Reviewer #3: 1. While the correlation between increased clag2 and growth is strong, the authors would ideally directly effect the role of clag2 expression level on growth/transport modulating expression via the glmS element introduced during the Nluc tagging (as long as glucosamine transport does not depend on PSAC).

The standard approach using Blasticidin-selectable plasmids may be complicated if there is a major change in blasticidin transport, which would need to be tested.

2. The mutational signature of the copy numbers suggest that amplification might have occurred in two steps, where the parental copy was amplified 8-10 fold, followed the acquisition of additional mutations one one of these copies, which was them amplified again. This is most apparent in the 3'UTR. This could be a sign that the additional mutations in clag2 are adaptive to growth in PGIM and leading to differential transport activities of the two sets of clag2 copies. Do any of these mutations result in coding changes? If not, this can be disregarded.

3. Figure 5C:re-check genomic copy number after tagging, as C3h-KO had 1 copy earlier and similar mRNA but the C3h-KO-Nluc has higher protein expression. CAS9 cleavage & repair at the locus could have led to changes in copy number.

4. Is CLAG2-dependent transport in C3h-KO-PGIM susceptible to inhibition by ISG-21 or residual transport inhibitors PRT1-4?

5. Can the fact that CLAG3-loss has a major effect on growth in PGIM but lesser effects on transport and lysis be explained be differences in the intracellular concentrations of Ile required for growth vs lysis?

**Part III – Minor Issues: Editorial and Data Presentation Modifications**

Reviewer #1: 1.Wording revisions:

Revise sentence on Page 4, lines 14–16 to something more cautious, e.g.:

"We examined sorbitol and isoleucine permeabilities in C3h-KO and KC5; while sorbitol is a well-established PSAC substrate, isoleucine uptake may involve additional or alternative transport pathways."

2.Suggestion for future experiments (optional, in discussion) to better understand the link between isoleucine uptake and direct PSAC-mediated transport, the authors could consider:

- Performing competition assays using sorbitol or other known PSAC substrates to determine if isoleucine uptake is competitively inhibited.

- Conducting transcriptomic analysis comparing WT and CLAG3-KO lines, with a focus on amino acid transporter expression.

- Generating and testing a PfAAT1 KO or knockdown strain in the clag3-KO background to assess compensatory or synergistic effects on isoleucine uptake.

3. Address clag2 pre-existence (Figure 4B):

The observed clag2 gene amplification under amino acid-limited conditions is a striking example of adaptive genome remodeling. As the authors note, this supports a broader model where nutrient stress, like drug pressure, can drive copy number variations to enhance parasite survival. This aligns with known mechanisms for resistance involving pfmdr1 or pfpm2/3. However, in Figure 4B, clag2 amplification may appear detectable in the C3h-KO parental line, (although not confirmed by WGS) suggesting that some degree of clag2 amplification may have pre-existed in the population, raising the possibility of selection on a pre-existing subpopulation rather than de novo amplification. The authors should discuss whether low-level pre-existing amplification may have been selected under PGIM pressure, or whether this reflects background noise. This clarification will help support the model of clag2-driven adaptation.

4. Figure S2A clarification:

In Figure S2A, the agarose gel analysis supports successful genome editing. However, assuming the primer pairs match those illustrated in Figure 5B, the p8 + p13 combination should, in principle, yield a smaller band in the KC5 line (wild-type for that locus). If this band is absent, please clarify whether this was expected or provide an explanation for the observed result.

5. Minor formatting consistency:

Ensure that acronyms (PGIM) are defined at first use.

Reviewer #2: -It is surprising that the copy number of clag2 increased up to 18-fold. Why are 18 copies of clag2 needed to compensate deletion of one copy of clag3? This massive increase may suggest that the CLAG2-based channel is less efficient than the CLAG3-based channel for the acquisition of the nutrients that are limiting in PGIM. An alternative possibility is that clag2 has a weaker promoter than clag3, which would imply that more copies of clag2 are needed to achieve the same transcript or protein levels. The latter possibility is supported by RNA-seq datasets available in PlasmoDB and several published RT-qPCR analyses. I recommend that the authors discuss these possible reasons for the unexpectedly high level of genetic amplification of clag2.

-In fig. 5A, data is presented in a way that does not enable comparison of clag3 transcript levels in the parental line with clag2 transcript levels in the KO, KO-PGIM and KO-rev lines (only values for each transcript relative to the parental line are shown). Please provide a figure (a new panel in Fig. 5 or a new supplementary figure) that enables this comparison, showing the transcript levels (relative to tubuline) for each clag gene in each parasite line. It would be interesting to see if total clag transcript levels increase in the KO-PGIM line or, in spite of massive clag2 amplification, total clag transcripts remain more or less constant, because of the weaker promoter of clag2.

-For a paper focused on increased clag2 expression as an adaptive mechanism, it would be relevant to discuss that clag2 was previously described as a clonally variant gene regulated by heterochromatin (at the epigenetic level), similar to clag3.1 and clag3.2 but unlike clag8 and clag9. This was demonstrated by transcriptional analysis of subclones that do or do not express the gene, and by genome-wide analysis of heterochromatin (a trait associated with clonally variant gene expression) showing that this epigenetic mark is present in clag2, clag3.1 and clag3.2 but not in clag8 or clag9. Clonally variant expression implies that the expression of this gene is dynamic and switching off the gene can provide a selective advantage under specific conditions, which fits with the observation that copy number decreased in the KO-rev line.

-Some statements in the manuscript may suggest that copy number variation is a usual clag genes regulation mechanism that the parasite can use in natural infections to adapt to changes in the host environment (e.g., a malnourished host). This is missleading. Copy number variation was observed in an artificial parasite line with no functional clag3 gene, a situation that does not occur in the field (no natural isolates without functional clag3 genes have so far been reported, to the best of my knowledge). Furthermore, large copy number variation has not been described in field isolates. It is also important to note that copy number variation cannot provide the rapid and dynamic plasticity needed for adaptation to the changing conditions of the human host. Therefore, in natural infections, regulation of clag3 genes most likely relies on epigenetic mechanisms, which are much faster and dynamic than genetic changes. Studying the adaptive alterations in the CLAG3-defficient line to grow in PGIM was instrumental to establish the role of CLAG2 in the formation of PSAC, but statements that misleadingly suggest that clags copy number variation may be a common mechanism of parasite adaptation should be modified (mainly end of Abstract and Author Summary, and second paragraph of the Discussion: p.13 line 10 and elsewhere).

-The results described in page 6 (Fig. 1E-G) indicate that the moderate decrease in permeability for the solutes tested in the KO line cannot account for the large growth defect of this parasite line in PGIM. The results presented afterwards do not provide an explanation for this intriguing observation, as increased CLAG2 levels restore permeability and largely revert the severe growth defect. Do the authors have an explanation, even if it is speculative, for why a small reduction in permeability in the KO line results in a severe growth defect? (larger than the effect on growth of a drug with a comparable impact on permeability). This should be discussed (in the Discussion).

-End of page 9 and elsewhere. Why do the authors call this observation “unexpected”? If the gene encoding CLAG3 has been disrupted, it is expected that a channel with the properties of a CLAG3-based PSAC cannot be present in the cells.

-Please indicated the position of the predicted TSSs for clag2 relative to the genetic amplification in Fig. 4A, using a zoomed-in view of the locus. The TSSs for most genes are described in several genome-wide datasets and available in PlasmoDB. This would give an idea of which is the functional unit amplified, presumably containing the full clag2 promoter.

-End of page 14. The data presented strongly suggest that different CLAGs confer different specificity and transport properties to the PSAC channel. Ref. 16 does not disprove this possibility, because it is possible that differences in nutrient permeability associated with different clag paralogs occur only for specific nutrients, which was not tested in ref 16. This should be made clear.

Reviewer #3: Does synchrony and stage of C3h-KO alter the fraction of "dormant" parasites that drive recrudescence when switched back to RPMI?

Figures would benefit from within-panel labels and have plenty of white space to fit them.

FigS2A: repeating the primer diagram would be helpful for parsing the figure.

FigS2B: Is this the expected localization pattern for clag2? Is there any localization to the RBC membrane ?

The discussion would be better off starting with the second paragraph as clag2 CNV is not an example of antigenic switching but rather a compensatory mechanism in response to the engineered loss of clag3 and selection with reduced nutrient serum, similar to other CNV that occur in response to drug selections.

Fig5C: Is the Nluc still active on the membrane even after being run on a denaturing gel? Or was this probed with anti-Nluc antibody? Non-experts in patch-clamping (like me) might benefit for a more context and elaboration of the results in Figure 3.

PLOS authors have the option to publish the peer review history of their article (what does this mean? ). If published, this will include your full peer review and any attached files.

**Do you want your identity to be public for this peer review?** For information about this choice, including consent withdrawal, please see our Privacy Policy .

Reviewer #1: No

Reviewer #2: No

Reviewer #3: **Yes: ** Björn Kafsack

**Figure resubmission:**
---

## [Editor Report · Decision Letter 1]

3 Oct 2025

Dear Dr. Desai,

We are pleased to inform you that your manuscript 'Nutrient stress dramatically increases malaria parasite clag2 copy number to increase host cell permeability and enable pathogen survival' has been provisionally accepted for publication in PLOS Pathogens.

Best regards,

Tania F. de Koning-Ward

Academic Editor

PLOS Pathogens

Dominique Soldati-Favre

Section Editor

PLOS Pathogens

Sumita Bhaduri-McIntosh

Editor-in-Chief

PLOS Pathogens

orcid.org/0000-0003-2946-9497

Michael Malim

Editor-in-Chief

PLOS Pathogens

orcid.org/0000-0002-7699-2064

Thank you for your detailed responses to the reviewers comments and for highlighting where the modifications have been made to the manuscript. These changes have greatly improved the manuscript and strenghthen the conclusions that have been drawn.
---

## [Editor Report · Acceptance letter]

Dear Dr. Desai,

We are delighted to inform you that your manuscript, "Nutrient stress dramatically increases malaria parasite clag2 copy number to increase host cell permeability and enable pathogen survival," has been formally accepted for publication in PLOS Pathogens.

Best regards,

Sumita Bhaduri-McIntosh

Editor-in-Chief

PLOS Pathogens

orcid.org/0000-0003-2946-9497

Michael Malim

Editor-in-Chief

PLOS Pathogens

orcid.org/0000-0002-7699-2064